# LoRA: Low-Rank Adaptation of Large Language Models

**Edward Hu**[*]    **Yelong Shen**[*]    **Phillip Wallis**    **Zeyuan Allen-Zhu**
**Yuanzhi Li**    **Shean Wang**    **Lu Wang**    **Weizhu Chen**
Microsoft Corporation
`edward.hu@mila.quebec`
`{yeshe, phwallis, zeyuana, swang, luw, wzchen}@microsoft.com`
`yuanzhil@andrew.cmu.edu`

## ABSTRACT

An important paradigm of natural language processing consists of large-scale pre-training on general domain data and adaptation to particular tasks or domains. As we pre-train larger models, full fine-tuning, which retrains all model parameters, becomes less feasible. Using GPT-3 175B as an example – deploying independent instances of fine-tuned models, each with 175B parameters, is prohibitively expensive. We propose **Lo**w-**R**ank **A**daptation, or LoRA, which freezes the pre-trained model weights and injects trainable rank decomposition matrices into each layer of the Transformer architecture, greatly reducing the number of trainable parameters for downstream tasks. Compared to GPT-3 175B fine-tuned with Adam, LoRA can reduce the number of trainable parameters by a factor of 10,000 and the GPU memory requirement by a factor of 3. LoRA performs on-par or better than fine-tuning in model quality on RoBERTa, DeBERTa, GPT-2, and GPT-3, despite having fewer trainable parameters, a higher training throughput, and, unlike adapters, *no additional inference latency*. We also provide an empirical investigation into rank-deficiency in language model adaptation, which sheds light on the efficacy of LoRA. We release a package that facilitates the integration of LoRA with PyTorch models and provide our implementations and model checkpoints for RoBERTa, DeBERTa, and GPT-2 at `https://github.com/microsoft/LoRA`.

## 1 INTRODUCTION

Many applications in natural language processing rely on adapting *one* large-scale, pre-trained language model to *multiple* downstream applications. Such adaptation is usually done via *fine-tuning*, which updates all the parameters of the pre-trained model. The major downside of fine-tuning is that the new model contains as many parameters as in the original model. As larger models are trained every few months, this changes from a mere "inconvenience" for GPT-2 (Radford et al., b) or RoBERTa large (Liu et al., 2019) to a critical deployment challenge for GPT-3 (Brown et al., 2020) with 175 billion trainable parameters.[1]

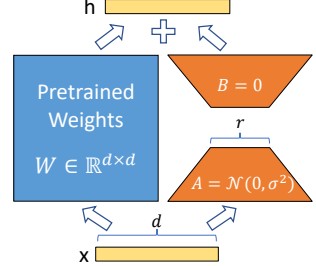

Many sought to mitigate this by adapting only some parameters or learning external modules for new tasks. This way, we only need to store and load a small number of task-specific parameters in addition to the pre-trained model for each task, greatly boosting the operational efficiency when deployed. However, existing techniques often introduce inference latency (Houlsby et al., 2019; Rebuffi et al., 2017) by extending model depth or reduce the model's usable sequence length (Li & Liang, 2021; Lester et al., 2021; Ham-

Figure 1: Our reparametrization. We only train $A$ and $B$.

---

[*]Equal contribution.

[1]While GPT-3 175B achieves non-trivial performance with few-shot learning, fine-tuning boosts its performance significantly as shown in Appendix A.

bardzumyan et al., 2020; Liu et al., 2021) (Section 3). More importantly, these method often fail to match the fine-tuning baselines, posing a trade-off between efficiency and model quality.

We take inspiration from Li et al. (2018a); Aghajanyan et al. (2020) which show that the learned over-parametrized models in fact reside on a low intrinsic dimension. We hypothesize that the change in weights during model adaptation also has a low "intrinsic rank", leading to our proposed **Lo**w-**R**ank **A**daptation (LoRA) approach. LoRA allows us to train some dense layers in a neural network indirectly by optimizing rank decomposition matrices of the dense layers' change during adaptation instead, while keeping the pre-trained weights frozen, as shown in Figure 1. Using GPT-3 175B as an example, we show that a very low rank (i.e., $r$ in Figure 1 can be one or two) suffices even when the full rank (i.e., $d$) is as high as 12,288, making LoRA both storage- and compute-efficient.

LoRA possesses several key advantages.

- A pre-trained model can be shared and used to build many small LoRA modules for different tasks. We can freeze the shared model and efficiently switch tasks by replacing the matrices $A$ and $B$ in Figure 1, reducing the storage requirement and task-switching overhead significantly.

- LoRA makes training more efficient and lowers the hardware barrier to entry by up to 3 times when using adaptive optimizers since we do not need to calculate the gradients or maintain the optimizer states for most parameters. Instead, we only optimize the injected, much smaller low-rank matrices.

- Our simple linear design allows us to merge the trainable matrices with the frozen weights when deployed, *introducing no inference latency* compared to a fully fine-tuned model, by construction.

- LoRA is orthogonal to many prior methods and can be combined with many of them, such as prefix-tuning. We provide an example in Appendix H.

**Terminologies and Conventions** We make frequent references to the Transformer architecture and use the conventional terminologies for its dimensions. We call the input and output dimension size of a Transformer layer $d_{model}$. We use $W_q$, $W_k$, $W_v$, and $W_o$ to refer to the query/key/value/output projection matrices in the self-attention module. $W$ or $W_0$ refers to a pre-trained weight matrix and $\Delta W$ its accumulated gradient update during adaptation. We use $r$ to denote the rank of a LoRA module. We follow the conventions set out by (Vaswani et al., 2017; Brown et al., 2020) and use Adam (Loshchilov & Hutter, 2019; Kingma & Ba, 2017) for model optimization and use a Transformer MLP feedforward dimension $d_{ffn} = 4 \times d_{model}$.

## 2 PROBLEM STATEMENT

While our proposal is agnostic to training objective, we focus on conditional language modeling as our motivating use case. Below is a brief description of the language modeling problem and, in particular, the maximization of conditional probabilities given a task-specific prompt.

Suppose we are given a pre-trained autoregressive language model $P_\Phi(y|x)$ parametrized by $\Phi$. For instance, $P_\Phi(y|x)$ can be a generic multi-task learner such as GPT (Radford et al., b; Brown et al., 2020) based on the Transformer architecture (Vaswani et al., 2017). Consider adapting this pre-trained model to downstream conditional text generation tasks, such as summarization, machine reading comprehension (MRC), and natural language to SQL (NL2SQL). Each downstream task is represented by a training dataset of context-target pairs: $\mathcal{Z} = \{(x_i, y_i)\}_{i=1,..,N}$, where both $x_i$ and $y_i$ are sequences of tokens. For example, in NL2SQL, $x_i$ is a natural language query and $y_i$ its corresponding SQL command; for summarization, $x_i$ is the content of an article and $y_i$ its summary.

During full fine-tuning, the model is initialized to pre-trained weights $\Phi_0$ and updated to $\Phi_0 + \Delta\Phi$ by repeatedly following the gradient to maximize the conditional language modeling objective:

$$\max_{\Phi} \sum_{(x,y)\in\mathcal{Z}} \sum_{t=1}^{|y|} \log\left(P_\Phi(y_t|x, y_{<t})\right) \tag{1}$$

One of the main drawbacks for full fine-tuning is that for *each* downstream task, we learn a *different* set of parameters $\Delta\Phi$ whose dimension $|\Delta\Phi|$ equals $|\Phi_0|$. Thus, if the pre-trained model is large (such as GPT-3 with $|\Phi_0| \approx 175$ Billion), storing and deploying many independent instances of fine-tuned models can be challenging, if at all feasible.

In this paper, we adopt a more parameter-efficient approach, where the task-specific parameter increment $\Delta\Phi = \Delta\Phi(\Theta)$ is further encoded by a much smaller-sized set of parameters $\Theta$ with $|\Theta| \ll |\Phi_0|$. The task of finding $\Delta\Phi$ thus becomes optimizing over $\Theta$:

$$\max_{\Theta} \sum_{(x,y)\in\mathcal{Z}} \sum_{t=1}^{|y|} \log\left(p_{\Phi_0+\Delta\Phi(\Theta)}(y_t|x, y_{<t})\right) \qquad (2)$$

In the subsequent sections, we propose to use a low-rank representation to encode $\Delta\Phi$ that is both compute- and memory-efficient. When the pre-trained model is GPT-3 175B, the number of trainable parameters $|\Theta|$ can be as small as $0.01\%$ of $|\Phi_0|$.

## 3 AREN'T EXISTING SOLUTIONS GOOD ENOUGH?

The problem we set out to tackle is by no means new. Since the inception of transfer learning, dozens of works have sought to make model adaptation more parameter- and compute-efficient. See Section 6 for a survey of some of the well-known works. Using language modeling as an example, there are two prominent strategies when it comes to efficient adaptations: adding adapter layers (Houlsby et al., 2019; Rebuffi et al., 2017; Pfeiffer et al., 2021; Rücklé et al., 2020) or optimizing some forms of the input layer activations (Li & Liang, 2021; Lester et al., 2021; Hambardzumyan et al., 2020; Liu et al., 2021). However, both strategies have their limitations, especially in a large-scale and latency-sensitive production scenario.

**Adapter Layers Introduce Inference Latency** There are many variants of adapters. We focus on the original design by Houlsby et al. (2019) which has two adapter layers per Transformer block and a more recent one by Lin et al. (2020) which has only one per block but with an additional LayerNorm (Ba et al., 2016). While one can reduce the overall latency by pruning layers or exploiting multi-task settings (Rücklé et al., 2020; Pfeiffer et al., 2021), there is no direct ways to bypass the extra compute in adapter layers. This seems like a non-issue since adapter layers are designed to have few parameters (sometimes <1% of the original model) by having a small bottleneck dimension, which limits the FLOPs they can add. However, large neural networks rely on hardware parallelism to keep the latency low, and adapter layers have to be processed sequentially. This makes a difference in the online inference setting where the batch size is typically as small as one. In a generic scenario without model parallelism, such as running inference on GPT-2 (Radford et al., b) medium on a single GPU, we see a noticeable increase in latency when using adapters, even with a very small bottleneck dimension (Table 1).

| Batch Size | 32 | 16 | 1 |
|---|---|---|---|
| Sequence Length | 512 | 256 | 128 |
| $|\Theta|$ | 0.5M | 11M | 11M |
| Fine-Tune/LoRA | 1449.4±0.8 | 338.0±0.6 | 19.8±2.7 |
| Adapter[L] | 1482.0±1.0 (+2.2%) | 354.8±0.5 (+5.0%) | 23.9±2.1 (+20.7%) |
| Adapter[H] | 1492.2±1.0 (+3.0%) | 366.3±0.5 (+8.4%) | 25.8±2.2 (+30.3%) |

Table 1: Infernece latency of a single forward pass in GPT-2 medium measured in milliseconds, averaged over 100 trials. We use an NVIDIA Quadro RTX8000. "$|\Theta|$" denotes the number of trainable parameters in adapter layers. Adapter[L] and Adapter[H] are two variants of adapter tuning, which we describe in Section 5.1. The inference latency introduced by adapter layers can be significant in an online, short-sequence-length scenario. See the full study in Appendix C.

This problem gets worse when we need to shard the model as done in Shoeybi et al. (2020); Lepikhin et al. (2020), because the additional depth requires more synchronous GPU operations such as `AllReduce` and `Broadcast`, unless we store the adapter parameters redundantly many times.

**Directly Optimizing the Prompt is Hard** The other direction, as exemplified by prefix tuning (Li & Liang, 2021), faces a different challenge. We observe that prefix tuning is difficult to optimize and that its performance changes non-monotonically in trainable parameters, confirming similar observations in the original paper. More fundamentally, reserving a part of the sequence length for adaptation necessarily reduces the sequence length available to process a downstream task, which we suspect makes tuning the prompt underperform other methods. We defer the study on task performance to Section 5.

## 4 OUR METHOD

We describe the simple design of LoRA and its practical benefits. The principles outlined here apply to any dense layers in deep learning models, though we only focus on certain weights in Transformer language models in our experiments as the motivating use case.

### 4.1 LOW-RANK-PARAMETRIZED UPDATE MATRICES

A neural network contains many dense layers which perform matrix multiplication. The weight matrices in these layers typically have full ranks. When adapting to a specific task, Aghajanyan et al. (2020) shows that the pre-trained language models have a low "instrisic dimension" and can still learn efficiently despite a random projection to a smaller subspace. Inspired by this, we hypothesize the updates to the weights also have a low "intrinsic rank" during adaptation. For a pre-trained weight matrix $W_0 \in \mathbb{R}^{d \times k}$, we constrain its update by representing the latter with a low-rank decomposition $W_0 + \Delta W = W_0 + BA$, where $B \in \mathbb{R}^{d \times r}, A \in \mathbb{R}^{r \times k}$, and the rank $r \ll \min(d, k)$. During training, $W_0$ is frozen and does not receive gradient updates, while $A$ and $B$ contain trainable parameters. Note both $W_0$ and $\Delta W = BA$ are multiplied with the same input, and their respective output vectors are summed coordinate-wise. For $h = W_0 x$, our modified forward pass yields:

$$h = W_0 x + \Delta W x = W_0 x + BAx \tag{3}$$

We illustrate our reparametrization in Figure 1. We use a random Gaussian initialization for $A$ and zero for $B$, so $\Delta W = BA$ is zero at the beginning of training. We then scale $\Delta W x$ by $\frac{\alpha}{r}$, where $\alpha$ is a constant in $r$. When optimizing with Adam, tuning $\alpha$ is roughly the same as tuning the learning rate if we scale the initialization appropriately. As a result, we simply set $\alpha$ to the first $r$ we try and do not tune it. This scaling helps to reduce the need to retune hyperparameters when we vary $r$ (Yang & Hu, 2021).

**A Generalization of Full Fine-tuning.** A more general form of fine-tuning allows the training of a subset of the pre-trained parameters. LoRA takes a step further and does not require the accumulated gradient update to weight matrices to have full ranks during adaptation. This means that when applying LoRA to all weight matrices and training all biases[2], we roughly recover the expressiveness of full fine-tuning by setting the LoRA rank $r$ to the rank of the pre-trained weight matrices. In other words, as we increase the number of trainable parameters [3], training LoRA roughly converges to training the original model, while adapter-based methods converges to an MLP and prefix-based methods to a model that cannot take long input sequences.

**No Additional Inference Latency.** When deployed in production, we can explicitly compute and store $W = W_0 + BA$ and perform inference as usual. Note that both $W_0$ and $BA$ are in $\mathbb{R}^{d \times k}$. When we need to switch to another downstream task, we can recover $W_0$ by subtracting $BA$ and then adding a different $B'A'$, a quick operation with very little memory overhead. Critically, this guarantees that we do not introduce any additional latency during inference compared to a fine-tuned model by construction.

### 4.2 APPLYING LORA TO TRANSFORMER

In principle, we can apply LoRA to any subset of weight matrices in a neural network to reduce the number of trainable parameters. In the Transformer architecture, there are four weight matrices in

---

[2]They represent a negligible number of parameters compared to weights.

[3]Inevitable when adapting to hard tasks.

the self-attention module ($W_q, W_k, W_v, W_o$) and two in the MLP module. We treat $W_q$ (or $W_k, W_v$) as a single matrix of dimension $d_{model} \times d_{model}$, even though the output dimension is usually sliced into attention heads. We limit our study to **only adapting the attention weights** for downstream tasks and freeze the MLP modules (so they are not trained in downstream tasks) both for simplicity and parameter-efficiency.We further study the effect on adapting different types of attention weight matrices in a Transformer in Section B.1. We leave the empirical investigation of adapting the MLP layers, LayerNorm layers, and biases to a future work.

**Practical Benefits and Limitations.** The most significant benefit comes from the reduction in memory and storage usage. For a large Transformer trained with Adam, we reduce that VRAM usage by up to $2/3$ if $r \ll d_{model}$ as we do not need to store the optimizer states for the frozen parameters. On GPT-3 175B, we reduce the VRAM consumption during training from 1.2TB to 350GB. With $r = 4$ and only the query and value projection matrices being adapted, the checkpoint size is reduced by roughly $10,000\times$ (from 350GB to 35MB)[4]. This allows us to train with significantly fewer GPUs and avoid I/O bottlenecks. Another benefit is that we can switch between tasks while deployed at a much lower cost by only swapping the LoRA weights as opposed to all the parameters. This allows for the creation of many customized models that can be swapped in and out on the fly on machines that store the pre-trained weights in VRAM. We also observe a 25% speedup during training on GPT-3 175B compared to full fine-tuning[5] as we do not need to calculate the gradient for the vast majority of the parameters. See Appendix L for how to calculate gradient when using LoRA.

LoRA also has its limitations. For example, it is not straightforward to batch inputs to different tasks with different $A$ and $B$ in a single forward pass, if one chooses to absorb $A$ and $B$ into $W$ to eliminate additional inference latency. Though it is possible to not merge the weights and dynamically choose the LoRA modules to use for samples in a batch for scenarios where latency is not critical.

## 5 EMPIRICAL EXPERIMENTS

We evaluate the downstream task performance of LoRA on RoBERTa (Liu et al., 2019), De-BERTa (He et al., 2021), and GPT-2 (Radford et al., b), before scaling up to GPT-3 175B (Brown et al., 2020). Our experiments cover a wide range of tasks, from natural language understanding (NLU) to generation (NLG). Specifically, we evaluate on the GLUE (Wang et al., 2019) benchmark for RoBERTa and DeBERTa. We follow the setup of Li & Liang (2021) on GPT-2 for a direct comparison and add WikiSQL (Zhong et al., 2017) (NL to SQL queries) and SAMSum (Gliwa et al., 2019) (conversation summarization) for large-scale experiments on GPT-3. See Appendix D for more details on the datasets we use. We use NVIDIA Tesla V100 for all experiments.

### 5.1 BASELINES

To compare with other baselines broadly, we replicate the setups used by prior work and reuse their reported numbers whenever possible. This, however, means that some baselines might only appear in certain experiments. We point out that for most tasks, our goal is not to compete with SOTA because 1) for tasks such as WikiSQL, SOTA uses specialized SQL-aware decoders, and; 2) SOTA makes frequent use of techniques such as ensembling and adversarial training, which might confound our study on adaptation. Thus, we focus on a scenario where we are given a powerful pre-trained autoregressive language model and use as few task-specific modifications as possible. See Appendix E for how we count the number of trainable parameters for different baselines.

**Fine-Tuning (FT)** is a common approach for adaptation. During fine-tuning, the model is initialized to the pre-trained weights and biases, and all model parameters undergo gradient updates.A simple variant is to update only some layers while freezing others. We include one such baseline reported in prior work (Li & Liang, 2021) on GPT-2, which adapts just the last two layers (**FTTop2**).

---

[4]We still need the 350GB model during deployment; however, storing 100 adapted models only requires 350GB + 35MB * 100 ≈ 354GB as opposed to 100 * 350GB ≈ 35TB.

[5]For GPT-3 175B, the training throughput for full fine-tuning is 32.5 tokens/s per V100 GPU; with the same number of weight shards for model parallelism, the throughput is 43.1 tokens/s per V100 GPU for LoRA.

| Model & Method | # Trainable Parameters | MNLI | SST-2 | MRPC | CoLA | QNLI | QQP | RTE | STS-B | Avg. |
|---|---|---|---|---|---|---|---|---|---|---|
| $\text{RoB}_{\text{base}}$ (FT)* | 125.0M | **87.6** | 94.8 | 90.2 | **63.6** | 92.8 | **91.9** | 78.7 | 91.2 | 86.4 |
| $\text{RoB}_{\text{base}}$ (BitFit)* | 0.1M | 84.7 | 93.7 | **92.7** | 62.0 | 91.8 | 84.0 | 81.5 | 90.8 | 85.2 |
| $\text{RoB}_{\text{base}}$ (Adpt$^D$)* | 0.3M | $87.1_{\pm.0}$ | $94.2_{\pm.1}$ | $88.5_{\pm1.1}$ | $60.8_{\pm.4}$ | $93.1_{\pm.1}$ | $90.2_{\pm.0}$ | $71.5_{\pm2.7}$ | $89.7_{\pm.3}$ | 84.4 |
| $\text{RoB}_{\text{base}}$ (Adpt$^D$)* | 0.9M | $87.3_{\pm.1}$ | $94.7_{\pm.3}$ | $88.4_{\pm.1}$ | $62.6_{\pm.9}$ | $93.0_{\pm.2}$ | $90.6_{\pm.0}$ | $75.9_{\pm2.2}$ | $90.3_{\pm.1}$ | 85.4 |
| $\text{RoB}_{\text{base}}$ (LoRA) | 0.3M | $87.5_{\pm.3}$ | $\mathbf{95.1}_{\pm.2}$ | $89.7_{\pm.7}$ | $63.4_{\pm1.2}$ | $\mathbf{93.3}_{\pm.3}$ | $90.8_{\pm.1}$ | $\mathbf{86.6}_{\pm.7}$ | $\mathbf{91.5}_{\pm.2}$ | **87.2** |
| $\text{RoB}_{\text{large}}$ (FT)* | 355.0M | 90.2 | **96.4** | **90.9** | 68.0 | 94.7 | **92.2** | 86.6 | 92.4 | 88.9 |
| $\text{RoB}_{\text{large}}$ (LoRA) | 0.8M | $\mathbf{90.6}_{\pm.2}$ | $96.2_{\pm.5}$ | $90.9_{\pm1.2}$ | $68.2_{\pm1.9}$ | $\mathbf{94.9}_{\pm.3}$ | $91.6_{\pm.1}$ | $\mathbf{87.4}_{\pm2.5}$ | $\mathbf{92.6}_{\pm.2}$ | **89.0** |
| $\text{RoB}_{\text{large}}$ (Adpt$^P$)† | 3.0M | $90.2_{\pm.3}$ | $96.1_{\pm.3}$ | $90.2_{\pm.7}$ | $\mathbf{68.3}_{\pm1.0}$ | $94.8_{\pm.2}$ | $\mathbf{91.9}_{\pm.1}$ | $83.8_{\pm2.9}$ | $92.1_{\pm.7}$ | 88.4 |
| $\text{RoB}_{\text{large}}$ (Adpt$^P$)† | 0.8M | $\mathbf{90.5}_{\pm.3}$ | $\mathbf{96.6}_{\pm.2}$ | $89.7_{\pm1.2}$ | $67.8_{\pm2.5}$ | $94.8_{\pm.3}$ | $91.7_{\pm.2}$ | $80.1_{\pm2.9}$ | $91.9_{\pm.4}$ | 87.9 |
| $\text{RoB}_{\text{large}}$ (Adpt$^H$)† | 6.0M | $89.9_{\pm.5}$ | $96.2_{\pm.3}$ | $88.7_{\pm2.9}$ | $66.5_{\pm4.4}$ | $94.7_{\pm.2}$ | $92.1_{\pm.1}$ | $83.4_{\pm1.1}$ | $91.0_{\pm1.7}$ | 87.8 |
| $\text{RoB}_{\text{large}}$ (Adpt$^H$)† | 0.8M | $90.3_{\pm.3}$ | $96.3_{\pm.5}$ | $87.7_{\pm1.7}$ | $66.3_{\pm2.0}$ | $94.7_{\pm.2}$ | $91.5_{\pm.1}$ | $72.9_{\pm2.9}$ | $91.5_{\pm.5}$ | 86.4 |
| $\text{RoB}_{\text{large}}$ (LoRA)† | 0.8M | $\mathbf{90.6}_{\pm.2}$ | $96.2_{\pm.5}$ | $90.2_{\pm1.0}$ | $68.2_{\pm1.9}$ | $94.8_{\pm.3}$ | $91.6_{\pm.2}$ | $85.2_{\pm1.1}$ | $92.3_{\pm.5}$ | 88.6 |
| $\text{DeB}_{\text{XXL}}$ (FT)* | 1500.0M | 91.8 | **97.2** | 92.0 | 72.0 | **96.0** | 92.7 | 93.9 | 92.9 | 91.1 |
| $\text{DeB}_{\text{XXL}}$ (LoRA) | 4.7M | $\mathbf{91.9}_{\pm.2}$ | $96.9_{\pm.2}$ | $\mathbf{92.6}_{\pm.6}$ | $\mathbf{72.4}_{\pm1.1}$ | $\mathbf{96.0}_{\pm.1}$ | $\mathbf{92.9}_{\pm.1}$ | $\mathbf{94.9}_{\pm.4}$ | $\mathbf{93.0}_{\pm.2}$ | **91.3** |

Table 2: RoBERTa$_{\text{base}}$, RoBERTa$_{\text{large}}$, and DeBERTa$_{\text{XXL}}$ with different adaptation methods on the GLUE benchmark. We report the overall (matched and mismatched) accuracy for MNLI, Matthew's correlation for CoLA, Pearson correlation for STS-B, and accuracy for other tasks. Higher is better for all metrics. * indicates numbers published in prior works. † indicates runs configured in a setup similar to Houlsby et al. (2019) for a fair comparison.

**Bias-only or BitFit** is a baseline where we only train the bias vectors while freezing everything else. Contemporarily, this baseline has also been studied by BitFit (Zaken et al., 2021).

**Prefix-embedding tuning (PreEmbed)** inserts special tokens among the input tokens. These special tokens have trainable word embeddings and are generally not in the model's vocabulary. Where to place such tokens can have an impact on performance. We focus on "prefixing", which prepends such tokens to the prompt, and "infixing", which appends to the prompt; both are discussed in Li & Liang (2021).

**Prefix-layer tuning (PreLayer)** is an extension to prefix-embedding tuning. Instead of just learning the word embeddings (or equivalently, the activations after the embedding layer) for some special tokens, we learn the activations after every Transformer layer. The activations computed from previous layers are simply replaced by trainable ones.

**Adapter tuning** as proposed in Houlsby et al. (2019) inserts adapter layers between the self-attention module (and the MLP module) and the subsequent residual connection. There are two fully connected layers with biases in an adapter layer with a nonlinearity in between. We call this original design **Adapter$^H$**. Recently, Lin et al. (2020) proposed a more efficient design with the adapter layer applied only after the MLP module and after a LayerNorm. We call it **Adapter$^L$**. This is very similar to another deign proposed in Pfeiffer et al. (2021), which we call **Adapter$^P$**. We also include another baseline call AdapterDrop (Rücklé et al., 2020) which drops some adapter layers for greater efficiency (**Adapter$^D$**). We cite numbers from prior works whenever possible to maximize the number of baselines we compare with; they are in rows with an asterisk (*) in the first column.

**LoRA** adds trainable pairs of rank decomposition matrices in parallel to existing weight matrices. As mentioned in Section 4.2, we only apply LoRA to $W_q$ and $W_v$ in most experiments for simplicity.

## 5.2 ROBERTA BASE/LARGE

RoBERTa (Liu et al., 2019) optimized the pre-training recipe originally proposed in BERT (Devlin et al., 2019a) and boosted the latter's task performance without introducing many more trainable parameters. While RoBERTa has been overtaken by much larger models on NLP leaderboards such as the GLUE benchmark (Wang et al., 2019) in recent years, it remains a competitive and popular pre-trained model for its size among practitioners. We take the pre-trained RoBERTa base (125M) and RoBERTa large (355M) from the HuggingFace Transformers library (Wolf et al., 2020)

| Model & Method | # Trainable Parameters | E2E NLG Challenge | | | | |
|---|---|---|---|---|---|---|
| | | BLEU | NIST | MET | ROUGE-L | CIDEr |
| GPT-2 M (FT)* | 354.92M | 68.2 | 8.62 | 46.2 | 71.0 | 2.47 |
| GPT-2 M (Adapter$^L$)* | 0.37M | 66.3 | 8.41 | 45.0 | 69.8 | 2.40 |
| GPT-2 M (Adapter$^L$)* | 11.09M | 68.9 | 8.71 | 46.1 | 71.3 | 2.47 |
| GPT-2 M (Adapter$^H$) | 11.09M | $67.3_{\pm.6}$ | $8.50_{\pm.07}$ | $46.0_{\pm.2}$ | $70.7_{\pm.2}$ | $2.44_{\pm.01}$ |
| GPT-2 M (FT$^{Top2}$)* | 25.19M | 68.1 | 8.59 | 46.0 | 70.8 | 2.41 |
| GPT-2 M (PreLayer)* | 0.35M | 69.7 | 8.81 | 46.1 | 71.4 | 2.49 |
| GPT-2 M (LoRA) | 0.35M | $\mathbf{70.4}_{\pm.1}$ | $\mathbf{8.85}_{\pm.02}$ | $\mathbf{46.8}_{\pm.2}$ | $\mathbf{71.8}_{\pm.1}$ | $\mathbf{2.53}_{\pm.02}$ |
| GPT-2 L (FT)* | 774.03M | 68.5 | 8.78 | 46.0 | 69.9 | 2.45 |
| GPT-2 L (Adapter$^L$) | 0.88M | $69.1_{\pm.1}$ | $8.68_{\pm.03}$ | $46.3_{\pm.0}$ | $71.4_{\pm.2}$ | $\mathbf{2.49}_{\pm.0}$ |
| GPT-2 L (Adapter$^L$) | 23.00M | $68.9_{\pm.3}$ | $8.70_{\pm.04}$ | $46.1_{\pm.1}$ | $71.3_{\pm.2}$ | $2.45_{\pm.02}$ |
| GPT-2 L (PreLayer)* | 0.77M | 70.3 | 8.85 | 46.2 | 71.7 | 2.47 |
| GPT-2 L (LoRA) | 0.77M | $\mathbf{70.4}_{\pm.1}$ | $\mathbf{8.89}_{\pm.02}$ | $\mathbf{46.8}_{\pm.2}$ | $\mathbf{72.0}_{\pm.2}$ | $2.47_{\pm.02}$ |

Table 3: GPT-2 medium (M) and large (L) with different adaptation methods on the E2E NLG Challenge. For all metrics, higher is better. LoRA outperforms several baselines with comparable or fewer trainable parameters. Confidence intervals are shown for experiments we ran. * indicates numbers published in prior works.

and evaluate the performance of different efficient adaptation approaches on tasks from the GLUE benchmark. We also replicate Houlsby et al. (2019) and Pfeiffer et al. (2021) according to their setup. To ensure a fair comparison, we make two crucial changes to how we evaluate LoRA when comparing with adapters. First, we use the same batch size for all tasks and use a sequence length of 128 to match the adapter baselines. Second, we initialize the model to the pre-trained model for MRPC, RTE, and STS-B, not a model already adapted to MNLI like the fine-tuning baseline. Runs following this more restricted setup from Houlsby et al. (2019) are labeled with †. The result is presented in Table 2 (Top Three Sections). See Section F.1 for details on the hyperparameters used.

## 5.3 DeBERTa XXL

DeBERTa (He et al., 2021) is a more recent variant of BERT that is trained on a much larger scale and performs very competitively on benchmarks such as GLUE (Wang et al., 2019) and SuperGLUE (Wang et al., 2020). We evaluate if LoRA can still match the performance of a fully fine-tuned DeBERTa XXL (1.5B) on GLUE. The result is presented in Table 2 (Bottom Section). See Section F.2 for details on the hyperparameters used.

## 5.4 GPT-2 medium/large

Having shown that LoRA can be a competitive alternative to full fine-tuning on NLU, we hope to answer if LoRA still prevails on NLG models, such as GPT-2 medium and large (Radford et al., b). We keep our setup as close as possible to Li & Liang (2021) for a direct comparison. Due to space constraint, we only present our result on E2E NLG Challenge (Table 3) in this section. See Section I.1 for results on WebNLG (Gardent et al., 2017) and DART (Nan et al., 2020). We include a list of the hyperparameters used in Section F.3.

## 5.5 Scaling up to GPT-3 175B

As a final stress test for LoRA, we scale up to GPT-3 with 175 billion parameters. Due to the high training cost, we only report the typical standard deviation for a given task over random seeds, as opposed to providing one for every entry. See Section F.4 for details on the hyperparameters used.

As shown in Table 4, LoRA matches or exceeds the fine-tuning baseline on all three datasets. Note that not all methods benefit monotonically from having more trainable parameters, as shown in Figure 2. We observe a significant performance drop when we use more than 256 special tokens for prefix-embedding tuning or more than 32 special tokens for prefix-layer tuning. This corroborates

| Model&Method | # Trainable Parameters | WikiSQL Acc. (%) | MNLI-m Acc. (%) | SAMSum R1/R2/RL |
|---|---|---|---|---|
| GPT-3 (FT) | 175,255.8M | **73.8** | 89.5 | 52.0/28.0/44.5 |
| GPT-3 (BitFit) | 14.2M | 71.3 | 91.0 | 51.3/27.4/43.5 |
| GPT-3 (PreEmbed) | 3.2M | 63.1 | 88.6 | 48.3/24.2/40.5 |
| GPT-3 (PreLayer) | 20.2M | 70.1 | 89.5 | 50.8/27.3/43.5 |
| GPT-3 (Adapter[H]) | 7.1M | 71.9 | 89.8 | 53.0/28.9/44.8 |
| GPT-3 (Adapter[H]) | 40.1M | 73.2 | **91.5** | 53.2/29.0/45.1 |
| GPT-3 (LoRA) | 4.7M | 73.4 | **91.7** | **53.8/29.8/45.9** |
| GPT-3 (LoRA) | 37.7M | **74.0** | 91.6 | 53.4/29.2/45.1 |

Table 4: Performance of different adaptation methods on GPT-3 175B. We report the logical form validation accuracy on WikiSQL, validation accuracy on MultiNLI-matched, and Rouge-1/2/L on SAMSum. LoRA performs better than prior approaches, including full fine-tuning. The results on WikiSQL have a fluctuation around $\pm 0.5\%$, MNLI-m around $\pm 0.1\%$, and SAMSum around $\pm 0.2/\pm 0.2/\pm 0.1$ for the three metrics.

similar observations in Li & Liang (2021). While a thorough investigation into this phenomenon is out-of-scope for this work, we suspect that having more special tokens causes the input distribution to shift further away from the pre-training data distribution. Separately, we investigate the performance of different adaptation approaches in the low-data regime in Section I.3.

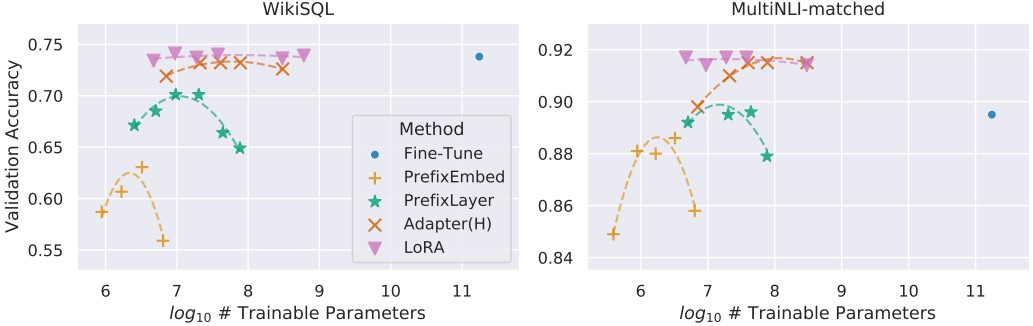

Figure 2: GPT-3 175B validation accuracy vs. number of trainable parameters of several adaptation methods on WikiSQL and MNLI-matched. LoRA exhibits better scalability and task performance. See Section I.2 for more details on the plotted data points.

## 6 RELATED WORKS

**Transformer Language Models.** Transformer (Vaswani et al., 2017) is a sequence-to-sequence architecture that makes heavy use of self-attention. Radford et al. (a) applied it to autoregressive language modeling by using a stack of Transformer decoders. Since then, Transformer-based language models have dominated NLP, achieving the state-of-the-art in many tasks. A new paradigm emerged with BERT (Devlin et al., 2019b) and GPT-2 (Radford et al., b) – both are large Transformer language models trained on a large amount of text – where fine-tuning on task-specific data after pre-training on general domain data provides a significant performance gain compared to training on task-specific data directly. Training larger Transformers generally results in better performance and remains an active research direction. GPT-3 (Brown et al., 2020) is the largest single Transformer language model trained to-date with 175B parameters.

**Prompt Engineering and Fine-Tuning.** While GPT-3 175B can adapt its behavior with just a few additional training examples, the result depends heavily on the input prompt (Brown et al., 2020). This necessitates an empirical art of composing and formatting the prompt to maximize a model's performance on a desired task, which is known as prompt engineering or prompt hacking.

Fine-tuning retrains a model pre-trained on general domains to a specific task Devlin et al. (2019b); Radford et al. (a). Variants of it include learning just a subset of the parameters Devlin et al. (2019b); Collobert & Weston (2008), yet practitioners often retrain all of them to maximize the downstream performance. However, the enormity of GPT-3 175B makes it challenging to perform fine-tuning in the usual way due to the large checkpoint it produces and the high hardware barrier to entry since it has the same memory footprint as pre-training.

**Parameter-Efficient Adaptation.** Many have proposed inserting *adapter* layers between existing layers in a neural network (Houlsby et al., 2019; Rebuffi et al., 2017; Lin et al., 2020). Our method uses a similar bottleneck structure to impose a low-rank constraint on the weight updates. The key functional difference is that our learned weights can be merged with the main weights during inference, thus not introducing any latency, which is not the case for the adapter layers (Section 3). A comtenporary extension of adapter is COMPACTER (Mahabadi et al., 2021), which essentially parametrizes the adapter layers using Kronecker products with some predetermined weight sharing scheme. Similarly, combining LoRA with other tensor product-based methods could potentially improve its parameter efficiency, which we leave to future work. More recently, many proposed optimizing the input word embeddings in lieu of fine-tuning, akin to a continuous and differentiable generalization of prompt engineering (Li & Liang, 2021; Lester et al., 2021; Hambardzumyan et al., 2020; Liu et al., 2021). We include comparisons with Li & Liang (2021) in our experiment section. However, this line of works can only scale up by using more special tokens in the prompt, which take up available sequence length for task tokens when positional embeddings are learned.

**Low-Rank Structures in Deep Learning.** Low-rank structure is very common in machine learning. A lot of machine learning problems have certain intrinsic low-rank structure (Li et al., 2016; Cai et al., 2010; Li et al., 2018b; Grasedyck et al., 2013). Moreover, it is known that for many deep learning tasks, a heavily over-parametrized neural network will exhibit low-rank properties after training (Oymak et al., 2019). Some prior works even explicitly impose the low-rank constraint when training the original neural network (Sainath et al., 2013; Povey et al., 2018; Zhang et al., 2014; Jaderberg et al., 2014; Zhao et al., 2016; Khodak et al., 2021; Denil et al., 2014); however, to the best of our knowledge, none of these works considers low-rank update to a frozen model for *adaptation to downstream tasks*. In theory literature, it is known that neural networks outperform other classical learning methods, including the corresponding (finite-width) neural tangent kernels (Allen-Zhu et al., 2019; Li & Liang, 2018) when the underlying concept class has certain low-rank structure (Ghorbani et al., 2020; Allen-Zhu & Li, 2019; Allen-Zhu & Li, 2020a). Another theory result in Allen-Zhu & Li (2020b) suggests that low-rank adaptations can be useful for adversarial training. In sum, we believe that our proposed low-rank adaptation update is well-motivated by the literature.

# 7 CONCLUSION AND FUTURE WORK

Fine-tuning enormous language models is prohibitively expensive in terms of the hardware required and the storage/switching cost for hosting independent instances for different tasks. We propose LoRA, an efficient adaptation strategy that neither introduces inference latency nor reduces input sequence length while retaining high model quality. Importantly, it allows for quick task-switching when deployed as a service by sharing the vast majority of the model parameters. While we focused on Transformer language models, the proposed principles are generally applicable to any neural networks with dense layers.

There are many directions for future works. 1) LoRA can be combined with other efficient adaptation methods, potentially providing orthogonal improvement. 2) The mechanism behind fine-tuning or LoRA is far from clear – how are features learned during pre-training transformed to do well on downstream tasks? We believe that LoRA makes it more tractable to answer this than full fine-tuning. 3) We mostly depend on heuristics to select the weight matrices to apply LoRA to. Are there more principled ways to do it? 4) Finally, the rank-deficiency of $\Delta W$ suggests that $W$ could be rank-deficient as well, which can inspire future works.

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
