# OpenReview forum: "LoRA: Low-Rank Adaptation of Large Language Models"
_ICLR.cc/2022/Conference — ICLR 2022 Poster_

### Official Review · Reviewer_jHjA · 2021-10-31

**Correctness:** 4
**Technical Novelty And Significance:** 3
**Empirical Novelty And Significance:** 3
**Recommendation:** 8
**Confidence:** 5

**Main Review:**

Strengths
- The idea of using this parallel low-ranked path to perform efficient fine-tuning is novel (to the best of my knowledge) and interesting. To elaborate, LoRA allows to simply add the low-ranked representation to the original weights, so to avoid further computation during inference, and this is possible since the model is trained in this way (by construction), check Eq 3. This is definitely a clever trick that allows not only an efficient inference but also efficient training since the optimizer (e.g., Adam) only need to store the gradients of the low-ranked parameters rather than the whole model (check the "Practical Benefits and Limitations." paragraph, Page 5).
- A large set of experiments (both in NLG and NLU) strongly supports the claim made by the authors.
- The paper is clearly motivated and covers a large amount of previous work. I especially enjoyed Section 3.

Minor weaknesses
- The training efficiency of LoRA is a bit unclear. In "Practical Benefits and Limitations." the authors mention a 2/3 reduction in VRAM by not storing the gradients of the previous steps in Adam, which is true but it is important to mention that the gradient of the original weights is still needed (although not stored) at the current optimization step to compute the gradient for the low-ranked weights (if I understood correctly your method). Anyhow, this step is needed also by other efficient fine-tuning methods (e.g. Adapter and Prompt-Tuning) and can be very costly in very large models (e.g., GPT-3). I would encourage the authors to provide a discussion about the training efficiency.
- The authors did not report any baselines outside of the efficient fine-tuning setting. This is important for the reader to understand how far an efficient model is compared to the current SOTA. By no means, the reader, and me, expect new SOTA, but it is good to have these references in the table.

**Summary Of The Paper:**

In this paper, the authors propose a LoRA a lightweight fine-tuning method for language models (LMs). LoRA achieves similar or better performance than full-model fine-tuning while adding only 0.01-0.05% of the original parameters and freezing the original weight of the LM, it is efficient in training and it does not add further computational cost during inference, as required by other methods.

The central idea is to only learn a parallel set of low-ranked parameters (Figure 1) rather than fine-tuning the entire model. To elaborate, the authors propose to 1) freeze the original LM, 2) create a parallel set of low-ranked parameters ($W=BA$ where $B\in\mathcal{R}^{d\times r}$ and $A\in\mathcal{R}^{r\times k}$) for every parameter in the original model, 3) training the model by only updating the low-ranked parameters where each hidden state is computed as $W_0 x + BA x$ (Eq. 3, $W_0\in\mathcal{R}^{d\times k}$ are the original frozen parameter of the model), and 4) test the model using the new set of parameters computed as $W_0 + BA$ (thus no additional computation at inference time).

The authors evaluate LoRA on both NLU (GLUE) and NLG (text-to-SQL, Dialogue Summary, E2E-NLG) tasks by comparing many models (Roberta, DeBerta, GPT-*) under different training settings and baselines such as full-finetuning, Bit-fit (only training bias vectors), prefix tuning methods, and Adapter methods. The results show that LoRA outperforms other efficient fine-tuning methods and often full-model finetuning, by only using a tiny number of parameters compared to the original model (Table 3/4). Moreover, LoRA does not add any inference overhead, as shown in Table 1, compared to other methods (e.g, Adapter), which required further computation.


**Summary Of The Review:**

The authors present an efficient fine-tuning method for LMs that freeze the original model and tune only an additional 0.01% of parameters. This method is efficient both in training and inference, and achieve comparable or better performance than full-model fine-tuning.

---

> ### Author Response · Authors · 2021-11-11
> **Response**
>
> Thank you Reviewer jHjA for reviewing our paper.
>
> > the authors mention a 2/3 reduction in VRAM by not storing the gradients of the previous steps in Adam, which is true but it is important to mention that the gradient of the original weights is still needed (although not stored) at the current optimization step to compute the gradient for the low-ranked weights (if I understood correctly your method).
>
> We would like to point out that the gradient of the pre-trained, frozen weights is not actually needed nor computed. Consider $h_i = W_i x_{i-1}$ and its LoRA counterpart $h_i = W_i x_{i-1} + BA x_{i-1}$. Given $\frac{\partial L}{\partial h_i}$, we would like to know $\frac{\partial L}{\partial B}$ and $\frac{\partial L}{\partial A}$. We denote $v = A x_{i-1}$.  $\frac{\partial L}{\partial B}$ is given by $\frac{\partial L}{\partial h_i} \otimes v$, and $\frac{\partial L}{\partial A}$ is given by $\frac{\partial L}{\partial v} \otimes x_{i-1}$. Note that we do not need to compute $\frac{\partial L}{\partial W_i}$ at any point, even though we compute $\frac{\partial L}{\partial x_{i-1}}$, which involves a matmul with $W_i$.
>
> > The authors did not report any baselines outside of the efficient fine-tuning setting. This is important for the reader to understand how far an efficient model is compared to the current SOTA. By no means, the reader, and me, expect new SOTA, but it is good to have these references in the table.
>
> Please see our response to common concerns. We would like to point out that we also report the full finetuning baseline, albeit without any advanced techniques such as ensembling or adversarial training, in addition to other efficient adaptation methods.

---

> > ### Comment · Reviewer_jHjA · 2021-11-11
> > **Re: Response**
> >
> > Thanks for the clarification about the gradient. I suggest adding (maybe in the appendix) these few steps which definitely helps to understand the efficiency of LoRA.
> >
> > About SOTA. I want to clarify that I personally do not expect to compare your results with SOTA models, but just to report it in the Table. This is useful for the reader to know what is (if any) the gap between efficient models (e.g. LoRA) and SOTA.

---

> > > ### Author Response · Authors · 2021-11-11
> > > **Response**
> > >
> > > Thank you for your prompt response!
> > >
> > > We will add the gradient derivation to our revision.
> > >
> > > We will also add SOTA numbers to our result and point out potential sources of the gap, if any.

---

### Official Review · Reviewer_TD94 · 2021-11-02

**Correctness:** 3
**Technical Novelty And Significance:** 2
**Empirical Novelty And Significance:** 3
**Recommendation:** 6
**Confidence:** 5

**Main Review:**

Strengths:
- Simple and efficient methods to utilize pre-trained models such as GPT-3 for model adaptation
- Good performance and comparison with fine-tuning baselines.

Weaknesses:
- The idea is incremental. The additional parameters have a similar intuition as adapters.
- Missing comparison to the SOTA models.
- The training efficiency remains unclear. It would be useful if the authors could report the time during training and compare the latency to the baseline models.


**Summary Of The Paper:**

The paper proposes a low-rank adaptation that compresses the pre-trained model weights by applying matrix decomposition (e.g., SVD) into large pre-trained models. The factorized pre-trained weights are frozen, and the small adaptation parameters are trainable during the finetuning stage. This method is very useful in the practical setting. They showed the effectiveness of applying this method in various downstream tasks by adding small adaptation parameters.

**Summary Of The Review:**

Justification:

I would say that the idea is relevant for achieving efficient pre-trained models that will be useful for many practical applications where latency is crucial. The paper is also well-written, with numerous experiments to support the claim. Therefore, I think the claim is supported with empirical results; however, the paper's novelty is limited. The time efficiency still needs to be supported by training and inference time information. The approach is incremental, very similar to Adapters; adding small trainable parameters to the model.

Typographical errors:
- comtenporary => contemporary

References related to Matrix decomposition by design (please consider to add):
- Compressing Pre-trained Language Models by Matrix Decomposition
- Greenformer: Factorization toolkit for efficient deep neural networks
- Factorization Tricks For LSTM Networks
- Lightweight and Efficient End-to-End Speech Recognition Using Low-Rank Transformer

Questions:
- What is the optimal rank for the factorized matrices? I was wondering to know if you have findings on choosing the optimal rank in the experiment.

I would be happy to increase my score if the authors address my concerns.

---

> ### Author Response · Authors · 2021-11-11
> **Response**
>
> Thank you Reviewer TD94 for reviewing our paper.
>
> > The idea is incremental. The additional parameters have a similar intuition as adapters… the paper's novelty is limited.
>
> We would like to point out a key advantage of our work over adapter is the option to eliminate inference latency, which is motivated by the practical challenge of deploying GPT-3 175B. This is enabled by a generalization of finetuning where we do not force the update to have full-rank. We argue that under this view, our method differs non-trivially from adapters, which learn external, sequential modules.
>
> In terms of novelty, while it’s true that low-rankness is commonly considered for model compression, it is not clear a priori if low-rank updates are sufficient to adapt a general-purpose LM to a specific task. The fact that full-rank finetuning is still the standard practice suggests that a simple method like LoRA was not expected to be sufficient. We argue that the simplicity of our design strengthens, rather than hurts, the novelty of our result.
>
> > Missing comparison to the SOTA models. The training efficiency remains unclear. It would be useful if the authors could report the time during training and compare the latency to the baseline models.
>
> Please see our response to common concerns.
>
> > What is the optimal rank for the factorized matrices? I was wondering to know if you have findings on choosing the optimal rank in the experiment.
>
> We study this in Appendix A.2. We apologize for the lack of pointers in the main text since we had to move the section (now Appendix A) to the appendix at the last minute due to space constraints. We will include a reference to that section in the main text in our revision.
>
>
> > Typographical errors:
> > comtenporary => contemporary
> > References related to Matrix decomposition by design (please consider to add):
> > Compressing Pre-trained Language Models by Matrix Decomposition
> > Greenformer: Factorization toolkit for efficient deep neural networks
> > Factorization Tricks For LSTM Networks
> > Lightweight and Efficient End-to-End Speech Recognition Using Low-Rank Transformer”
>
> Thank you for your suggestions. We will incorporate them in our revision.
>
> Please consider raising your score if we are able to address some of your concerns.

---

> > ### Comment · Reviewer_TD94 · 2021-11-17
> > **Response**
> >
> > Thanks for the clarification. I have changed my score.

---

### Official Review · Reviewer_i5pN · 2021-11-03

**Correctness:** 4
**Technical Novelty And Significance:** 2
**Empirical Novelty And Significance:** 3
**Recommendation:** 8
**Confidence:** 4

**Main Review:**

This paper is well motivated and easy to follow. Although the idea of using low-rank approximation of neural networks is not new, this work is the first to propose adding low-rank matrices as the task-specific parameters in the pretrained models. The benefits of this approach are obvious. It introduces no additional computation in single-task inference compared to adapters and prompt-based methods, and is much more training efficient than full model finetuning. Since the method works for any parameter matrix, it can be generally applicable to most of the neural network models. Experiments also demonstrate its efficacy on several pretrained language models and on both NLU and NLG tasks.

One limitation of this method (also mentioned in the paper) is how the model works for multi-task serving setting, which is possibly the most common scenario in deploying large-scale pretrained models. Since LoRA is able to meet this requirement by not merging the pretrained weights (W_0) and low-rank weights (BA), it will be interesting if the author can report the comparison results of inference latency among (no-merging) LoRA and other methods. Moreover, with regard to training efficiency, it is better to compare with adapters and prompt/prefix-tuning with the same amount of trainable parameters. Another line of work on mixture of experts (MoE) also shows promising results on mult-task/domain training (e.g., DEMix Layers: https://arxiv.org/abs/2108.05036), where we can selectively train expert modules for specific downstream task (while freezing the rest parameters). It will be interesting to see how RoLA compares with it.


**Summary Of The Paper:**

This paper proposes a low-rank adaptation method LoRA for application of pretrained models for downstream tasks. Each downstream task is associated with a set of trainable low-rank matrices that can be added to the pretrained parameters. Since the amount of low-rank parameters are orders of magnitude smaller than that of the pretrained parameters, the finetuning cost as well as the inference latency of LoRA is lower than existing methods including model finetuning, adpaters, and prompt tuning. Experiments on several pretrained models (RoBERTa, DeBERTa, GPT-2, GPT-3) show that RoLA achieves comparable or even better performances than existing methods.
The proposed method is simple and effective in adapting pretrained models to downstream tasks, which could become one of the major solutions among the finetuning, adapters and promp-tuning. The experiments are comprehensive, showing that LoRA works well in different types of pretrained models and tasks.


**Summary Of The Review:**

This work proposes to add parameters of low-rank matrices into pretrained model, which are tuned for specific downstream task. The method is effective under different types of pretrained models and tasks, and benefits the deployment of large-scale models. My overall recommendation to this paper is accept.

---

> ### Author Response · Authors · 2021-11-11
> **Response**
>
> Thank you Reviewer i5pN for reviewing our paper.
>
> > Since LoRA is able to meet this requirement by not merging the pretrained weights (W_0) and low-rank weights (BA), it will be interesting if the author can report the comparison results of inference latency among (no-merging) LoRA and other methods.
>
> That’s a good suggestion. We will say more about this in our revision. So far, our understanding is that for the same number of trainable parameters, the latency of unmerged LoRA is similar to but does not exceed that of Adapters, because it’s still possible to parallelize the extra work introduced by LoRA, but the same isn’t true for Adapters.
>
> > Moreover, with regard to training efficiency, it is better to compare with adapters and prompt/prefix-tuning with the same amount of trainable parameters.
>
> We used the same number of trainable parameters for LoRA and other baselines for NLU and GPT-2, though not for GPT-3 for the following reasons: 1) adapters include biases, which means it has more trainable parameters than LoRA even when both have a bottleneck dim r=1; this explains the difference between the 4.7M LoRA baseline and the 7.1M Adapter baseline. We try to even them out when we scale up to ~40M params. 2) the performance of prefix-tuning does not improve monotonically in the number of trainable params, as shown in Figure 2. As a result, the Pre-Emb baseline only has 3.2M params since the performance will be worse if we use more parameters. Hope this answers your question!
>
>
> > Another line of work on mixture of experts (MoE) also shows promising results on mult-task/domain training (e.g., DEMix Layers: https://arxiv.org/abs/2108.05036), where we can selectively train expert modules for specific downstream task (while freezing the rest parameters). It will be interesting to see how RoLA compares with it.
>
> This is quite interesting. We in fact have on-going work exploring the connection between MoE and low-rank updates. Please stay tuned!

---

### Official Review · Reviewer_Vqdv · 2021-11-03

**Correctness:** 2
**Technical Novelty And Significance:** 3
**Empirical Novelty And Significance:** 2
**Recommendation:** 6
**Confidence:** 4

**Main Review:**

I think the paper presents an interesting idea and has the potential to be impactful. But its execution can be significantly improved:
- The paper overclaims the benefit of low-rank finetuning without enough experiments to back it up.  LoRA only finetunes W_q and W_v. One missing baseline is a full-rank finetuning model, which only updates W_q and W_v. The memory/parameter efficiency of LoRA is exaggerated.
- Thorough proof reading is needed. Please see below for a long, but by no means exhaustive list of typos.-
- One may argue that LoRA can take more training steps to converge due to low-rank updates. So it would be great to compare learning curves and training time overhead.

Details:
- What does VRAM stand for? If it is GPU memory, then the 2/3 saving of training memory is quite surprising to me—LoRA still needs to build the forward and backward computation graphs for the whole model. Could the authors clarify?
- A (probably more general) variant of LoRA would be, instead of learning A and B, one forces the gradients to be of low-rank. So accumulating multiple low-rank steps could still yield a higher-rank update to the parameters. Have the authors considered this?

Typos, writing suggestions:
- “Reduce something by 10K/3 times” is strange. Please consider changing it to reduce by a factor of 10K, or reduce by 2/3
- Please P(X), P(X = x), and p(x).
- Whenever the paper talks about autoregressive language models, the math is never a language model.
- A non-issue -> not an issue
- Less performant -> underperform
- These layers have full-rank -> these layer have full ranks.
- An inevitability -> Inevitable

**Summary Of The Paper:**

This paper presents LoRA, a method aiming to improve the efficiency of fine-tuning large language models. Specifically, it fixes the underlying pretrained model, and learn low-rank (by construction) parameter matrices to update the model. In this way, the amount of parameters to finetune is significantly reduced if the rank is small. Compared to other finetuning methods, LoRA performs similarly or better, with improved memory and parameter efficiency.


**Summary Of The Review:**

Se main review

---

> ### Author Response · Authors · 2021-11-11
> **Response**
>
> Thank you reviewer Vqdv for reviewing our paper.
>
> > The paper overclaims the benefit of low-rank finetuning without enough experiments to back it up… The memory/parameter efficiency of LoRA is exaggerated.
>
> We are sorry that you felt like we didn’t present our result appropriately. In the abstract, we claim that “Compared to GPT-3 175B fine-tuned with Adam, LoRA can reduce the number of trainable parameters by 10,000 times and the GPU memory requirement by 3 times,” which reflects what we see in practice. We elaborate on this claim in Section 4.2 under “Practical Benefits and Limitations.” Of course, we do not achieve this kind of extreme parameter efficiency on much smaller models. Please let us know which claim in particular isn’t well supported in your opinion, and we will be happy to revise accordingly.
>
> > LoRA only finetunes W_q and W_v. One missing baseline is a full-rank finetuning model, which only updates W_q and W_v.
>
> We include such a baseline in Appendix I.2 (see r=1024), which shows that full-rank updates to just W_q and W_v actually underperform low-rank updates. We will add a reference to it in the main text. We did not extend this baseline to other settings such as RoBERTa and GPT-3 due to the limited budget for experiments and the fact that most practitioners just finetune all the weights.
>
> > One may argue that LoRA can take more training steps to converge due to low-rank updates. So it would be great to compare learning curves and training time overhead.
>
> Please see our response to common concerns.
>
> > What does VRAM stand for? If it is GPU memory, then the 2/3 saving of training memory is quite surprising to me—LoRA still needs to build the forward and backward computation graphs for the whole model. Could the authors clarify?
>
> Yes, we will clarify in our revision that “VRAM” refers to “Video RAM” or “GPU memory.” We elaborate on this claim in Section 4.2 under “Practical Benefits and Limitations.” The short answer is that we do not need to store the adaptive optimizer states (e.g., momentum and second order moment for Adam) for the pretrained parameters. This represents the upper-bound of potential memory savings; in practice, we approach this upper-bound on GPT-3 175B, our motivating use case. Note that this upper-bound is achieved when VRAM usage is dominated by weights and optimizer states. For smaller models where VRAM is dominated by activations, the savings will be less.
>
>
> > A (probably more general) variant of LoRA would be, instead of learning A and B, one forces the gradients to be of low-rank. So accumulating multiple low-rank steps could still yield a higher-rank update to the parameters. Have the authors considered this?
>
> If we use SGD for finetuning, the gradient contribution to a weight matrix $W_i$ from a single token will be a rank-1 matrix since it’s the result of an outer-product between the gradient of $h_i$ (the pre-activation after $W_i$) and $x_{i-1}$ (the activation before $W_i$). Indeed, when we accumulate the contributions from many tokens across batches and training steps, we derive a high-rank (and most likely, full-rank) update to the parameters. This takes us back to finetuning, since the “number of updates * number of tokens per sequence * number of sequences per batch” is usually much larger than the full rank of a weight matrix. We are considering a setup where we periodically prune the rank after accumulating some gradient updates, akin to the Frank-Wolfe method; however, this introduces a lot of complexity and is finicky without major payoffs in most applications. As a result, we choose to keep our presentation simple in this work.
>
>
> > Typos, writing suggestions:
> > Reduce something by 10K/3 times” is strange. Please consider changing it to reduce by a factor of 10K, or reduce by 2/3
> > Please P(X), P(X = x), and p(x).
> > Whenever the paper talks about autoregressive language models, the math is never a language model.
> > A non-issue -> not an issue
> > Less performant -> underperform
> > These layers have full-rank -> these layer have full ranks.
> > An inevitability -> Inevitable”
>
> Thank you for the detailed suggestions. We will incorporate them in our revision.
>
> In addition, we would like to point out that a key advantage for LoRA compared to adapters is that it does not always introduce inference latency, which is important for latency sensitive scenarios such as online-inferencing with GPT-3 175B.
>
> Please consider raising your score if we are able to address some of your concerns.

---

> > ### Comment · Reviewer_Vqdv · 2021-11-21
> > **Thanks for the response**
> >
> > Thank the authors for the response, which addressed some of my concerns. There are two remaining:
> > - Thanks for the pointer to I.2. However, it is not clear to me how this can be seen as a finetuning baseline that only updates $W_q$ and $W_v$. The baseline I would like to see is a standard finetuning model, but only updates $W_q$ and $W_v$. It does not use the $\Delta W=AB$ parameterization. One may argue that if $r\ll d, k$, then the  $\Delta W=AB$ parameterization is equivalent to standard finetuning. Be that as it may at test time, this is not the case at training time. The former (1) has a very different training dynamic, (2) can be drastically overparameterized and contributes to overfitting. Tables 5 and 6 present similar results, but do not include the baseline I suggested either. Please let me know if this one is already in the paper/appendix but I missed it.
> > - On memory saving. Thanks for the clarification on memory overhead! It is definitely interesting to know that Adam's momentum terms have such a huge impact on memory. I want to follow up with several questions/suggestions. If one uses non-momentum optimizers (e.g., SGD, Adadelta), is it correct that the training memory saving would be much less than 2/3, if there is any at all? The memory wording in the abstract might confuse readers that the memory saving holds at test time (which is not the case); the wording at the end of the intro is pretty good. I suggest, maybe in 4.2, explicitly saying that the memory saving is training only and does not translate to evaluation.

---

> > > ### Author Response · Authors · 2021-11-22
> > > **Response**
> > >
> > > Thank you for the clarification. You are right that I.2 (r=1024) does not exactly correspond to finetuning just $W_q$ and $W_v$.  It just shows that tuning just $W_q$ and $W_v$ with high-rank updates doesn't significantly outperform (it in fact underperforms) that with low-rank updates. We will look into prior works for such a baseline and consider running it before camera-ready if none is available for the reasons you have pointed out.
> > >
> > > - If one uses non-momentum optimizers (e.g., SGD, Adadelta), is it correct that the training memory saving would be much less than 2/3, if there is any at all?
> > >
> > > The memory saving depends on what the optimizer needs to store during training. For example, SGD without momentum doesn't need to store anything, therefore LoRA won't change how much VRAM the optimizer consumes. Though we still save some compared to full finetuning since the gradient for pretrained parameters doesn't need to be instantiated. The exact saving will depend on the implementation. If we naively calculate all the gradient matrices first, which is what happens if one calls `loss.backward()` in Pytorch (if my understanding is correct), the saving can still be as high as 50%; if one uses and discards the gradient immediately after its calculation, the memory saving will be at most $\frac{1}{L}$ where $L$ is the number of layers.
> > >
> > > Adadelta, similar to Adam, requires storing two states per parameter ($E[g^2]$ and $E[{\Delta x}^2]$ in the original paper, Algorithm 1). Our claim for Adam will hold in this case.
> > >
> > > Thanks for pointing out the ambiguity regarding training vs. inference. We will add "during training" to our claim of memory saving in the abstract.

---

> > > > ### Comment · Reviewer_Vqdv · 2021-11-24
> > > > **Thanks for the clarification!**
> > > >
> > > > Thank you for the clarification on the optimizers' implementations and how they are handled in PyTorch. Here are my thoughts on the current version:
> > > > - I still like the idea a lot, and do think it has the potential. But most of the claims and empirical gains are highly confounded with optimizer and auto-diff toolkit. This is probably unavoidable for a paper focusing on efficiency, since it is very hard to disentangle efficiency gain from the underlying machine learning systems. In this regard, I think the paper can be improved if it can (1) include some theoretical discussion on efficiency, and (2) honestly discuss how the savings by LoRA may come down to the toolkit and optimizer, just as in the response, but with more empirical evidence and less hedges.
> > > > - Finetuning only $W_q$ and $W_V$ is perhaps the most important baseline to LoRA, but it is missing. It cannot be simulated by any existing setting in the paper (see earlier this thread). Intuitively, I would expect that this baseline could achieve similar efficiency improvements and accuracy to LoRA. If it is the case, most of the paper's claims need to be revised. It is too late for the authors to update the paper. But if, by any chance, the authors are able to get convincing results of this baseline during the discussion period (by either running the experiments or citing previous works), I will be happy to revise my score.
> > > >
> > > > At this point, I keep my score unchanged. Although I'd like to increase it to 4 since my concern on training efficiency is addressed, the system does not give me this option.

---

> > > > > ### Author Response · Authors · 2021-11-30
> > > > > **Response**
> > > > >
> > > > > - I still like the idea a lot, and do think it has the potential. But most of the claims and empirical gains are highly confounded with optimizer and auto-diff toolkit. This is probably unavoidable for a paper focusing on efficiency, since it is very hard to disentangle efficiency gain from the underlying machine learning systems. In this regard, I think the paper can be improved if it can (1) include some theoretical discussion on efficiency, and (2) honestly discuss how the savings by LoRA may come down to the toolkit and optimizer, just as in the response, but with more empirical evidence and less hedges.
> > > > >
> > > > > We will improve our presentation in our revision regarding how our VRAM savings depends on the optimizer used (though we already noted throughout that we use Adam) and the production environment, e.g., if VRAM is dominated by model weights. However, we would like to point out that other efficiency gains, such as the reduction in trainable parameters (smaller ckpts and faster task-switching) and speedup during training, are not as system-dependent. We also disagree that our claim is "highly confounded with auto-diff toolkit," since gradient-based learning is fairly fundamental to how DL is done today and all toolkits perform roughly the same thing in that regard. We did digress slightly in our previous response when we discussed potentially more memory savings by calculating gradients sequentially instead of all at once, but that was pedagogical in nature and shouldn't be confused with the claim we made in the paper, which we have kept simple and true to what we see in practice.
> > > > >
> > > > > - Finetuning only W_q and W_v is perhaps the most important baseline to LoRA, but it is missing. It cannot be simulated by any existing setting in the paper (see earlier this thread). Intuitively, I would expect that this baseline could achieve similar efficiency improvements and accuracy to LoRA. If it is the case, most of the paper's claims need to be revised. It is too late for the authors to update the paper. But if, by any chance, the authors are able to get convincing results of this baseline during the discussion period (by either running the experiments or citing previous works), I will be happy to revise my score.
> > > > >
> > > > > We disagree that finetuning only $W_q$ and $W_v$ can achieve an efficiency improvement close to that of LoRA, even if it might obtain comparable accuracy (see our additional experiments in the next paragraph). The majority of the reduction in parameters comes from parametrizing the dense update matrix with a pair of low-rank matrices. For a square matrix of $d\times d$ and low-rank bottleneck $r$, we can expect a reduction in parameters of magnitude $\mu=\frac{d}{2\times r}$. For GPT-2 M, $\mu=128$ while the efficiency gain from tuning just $W_q$ and $W_v$ is $6$. For GPT-3 175B, $\mu$ can be as large as 3072 (r=2) or 6144 (r=1) while the efficiency gain from tuning just $W_q$ and $W_v$ is still $6$.
> > > > >
> > > > > With that said, we ran some preliminary experiments on GPT-2 M where we only finetune attention weights while keeping others frozen.
> > > > >
> > > > > | E2E          | params        | BLEU  | NIST | MET   | ROUGE-L | CIDEr |
> > > > > |--------------|---------------|-------|------|-------|---------|-------|
> > > > > | FT full      | 354.8M (100%) | 68.20 | 8.62 | 46.20 | 71.00   | 2.47  |
> > > > > | FT attn only | 75.5M (21%)   | 67.19 | 8.49 | 45.88 | 70.71   | 2.43  |
> > > > > | LoRA         | 0.35M (0.1%)  | 70.40 | 8.85 | 46.80 | 71.80   | 2.53  |
> > > > >
> > > > > |              | BLEU          |       |       | MET   |      |      | TER ↓ |      |      |      |
> > > > > |--------------|---------------|-------|-------|-------|------|------|-------|------|------|------|
> > > > > | WebNLG       | params        | U     | S     | A     | U    | S    | A     | U    | S    | A    |
> > > > > | FT full      | 354.8M (100%) | 27.70 | 64.20 | 46.50 | 0.30 | 0.45 | 0.38  | 0.76 | 0.33 | 0.53 |
> > > > > | FT attn only | 75.5M (21%)   | 44.26 | 62.86 | 54.42 | 0.37 | 0.45 | 0.41  | 0.49 | 0.33 | 0.41 |
> > > > > | LoRA         | 0.35M (0.1%)  | 46.70 | 62.10 | 55.30 | 0.38 | 0.44 | 0.41  | 0.46 | 0.33 | 0.39 |
> > > > >
> > > > > | DART         | params        | BLEU  | MET   | TER ↓ |
> > > > > |--------------|---------------|-------|-------|-------|
> > > > > | FT full      | 354.8M (100%) | 46.20 | 0.39  | 0.46  |
> > > > > | FT attn only | 75.5M (21%)   | 47.58 | 0.40  | 0.47  |
> > > > > | LoRA         | 0.35M (0.1%)  | 47.10 | 0.39  | 0.46  |
> > > > >
> > > > > Overall, the task performance result is mixed on WebNLG and DART, while LoRA has a small lead on E2E. However, as noted above, finetuning just the attention weights isn't nearly as parameter-efficient as LoRA, which means larger model checkpoints and more task-switching overhead.
> > > > >
> > > > > Edit: fixed the formatting of the last table

---

> > > > > > ### Comment · Reviewer_Vqdv · 2021-11-30
> > > > > > **Re: Response**
> > > > > >
> > > > > > Thank you for the efforts of conducting additional experiments! I have increased my score.
> > > > > >
> > > > > > I encourage the authors to include these results in the revision, and adjust the claims to more accurately reflect the savings from (a) low-rank updates and (b) finetuning a subset of the parameters.

---

### Author Response · Authors · 2021-11-11
**Response to common concerns**

We thank all of the reviewers for your thoughtful reviews.
Here we address some common concerns among all reviewers.

 > The training efficiency of LoRA isn’t clear. A comparison of training time and learning curves would be helpful.

Thank you for the suggestion. We will include a comparison of learning curves in our revision. Attached are the learning curves from GPT-3 175B on WikiSQL. We picked GPT-3 because we don’t have direct access to the finetuning learning curves for NLU and GPT-2 since we cited published results.

https://pasteboard.co/QjDls2dAWaHX.png (We note that LoRA actually reaches the best validation loss slightly faster than finetuning and adapter.)

In terms of training time, as described in footnote 5 on page 5, we see a higher training throughput compared to full finetuning, which translates to a shorter overall training time since we fixed the total number of updates.


 > There is no comparison to SOTA models on the tasks reported.

Here we assume SOTA means the best numbers reported on a given dataset. We have shown that LoRA can achieve comparable or better numbers than finetuning on several benchmarks. However, we understand that some orthogonal techniques such as adversarial training, model ensemble, and data augmentation, which are often combined to achieve SOTA runs, could further improve our numbers. We hope to explore their complementarity with LoRA in a future work since it's nontrivial to apply these techniques in a fair comparison. Nonetheless, we welcome suggestions from reviewers regarding what other baselines we should include to strengthen our paper.

Edit: changed the image host to one (hopefully) with fewer ads; made our response to the 2nd concern more succinct.

---

### Decision · Program_Chairs · 2022-01-20

**Decision:**

Accept (Poster)

**Comment:**

This paper introduces a new method for fine-tuning large language models, which is lightweight since it only adds a small amount of parameters, while keeping the original parameters frozen. The main idea is to add a low rank matrix which is learned during fine-tuning to the original weight matrices of the model, which are frozen. The reviewers agreed that the method is simple, original and well motivated. Moreover, it compares well compared to other fine-tuning baselines, such as adaptors or full fine-tuning. For these reasons, I recommend to accept this work to the ICLR conference.